# A Retrospective on the Innovative Sustainable Valorization of Cereal Bran in the Context of Circular Bioeconomy Innovations

Tabussam Tufail [1,*], Huma Bader Ul Ain [1], Farhan Saeed [2], Makia Nasir [3], Shahnai Basharat [1], Mahwish [4], Alexandru Vasile Rusu [5,6], Muzzamal Hussain [2], João Miguel Rocha [7,8], Monica Trif [9,*] and Rana Muhammad Aadil [10]

1   University Institute of Diet and Nutritional Sciences, Faculty of Allied Health Sciences, The University of Lahore, Main Campus, Lahore 54000, Pakistan
2   Department of Food Science, Government College University, Faisalabad 38000, Pakistan
3   Physio College of Rehabilitation Sciences, Multan 60650, Pakistan
4   Institute of Home Science, University of Agriculture, Faisalabad 38000, Pakistan
5   Life Science Institute, University of Agricultural Sciences and Veterinary Medicine of Cluj-Napoca, 400372 Cluj-Napoca, Romania
6   Faculty of Animal Science and Biotechnology, University of Agricultural Sciences and Veterinary Medicine of Cluj-Napoca, 400372 Cluj-Napoca, Romania
7   Laboratory for Process Engineering, Environment, Biotechnology and Energy, Faculty of Engineering, University of Porto, 4099-002 Porto, Portugal
8   Associate Laboratory in Chemical Engineering, Faculty of Engineering, University of Porto, 4099-002 Porto, Portugal
9   Food Research Department, Centre for Innovative Process Engineering (CENTIV) GmbH, 28857 Syke, Germany
10  National Institute of Food Science and Technology, University of Agriculture, Faisalabad 38000, Pakistan
*   Correspondence: tabussam.tufail@dnsc.uol.edu.pk (T.T.); monica_trif@hotmail.com (M.T.)

**Abstract:** Handling industrial agricultural wastes is a requirement for industrial waste management in the context of circular bioeconomy innovations. The recovery and re-use of agricultural wastes and their by-products have become an important topic of research and development to investigate their functional and nutraceutical properties. The bioeconomy provides an opportunity to create innovative bio-based products and processes, thereby opening up new markets. Agricultural waste contains a high concentration of protein, fat, carbohydrates, fiber, and other functional compounds such as antioxidants, which can be used to add value to a variety of food products. Due to its higher nutritional profile, cereal bran, as an agricultural waste and by-product, has a variety of functional and nutraceutical properties. Despite the fact that it is rich in bioactive compounds with health benefits, cereal bran is still underutilized in the food system. It can be used either directly for the processing of various foods or the extraction of various bioactive components present therein. Furthermore, the extracts from cereal bran have been used to obtain antioxidants, antibiotics, vitamins, and enzymes as functional components to be employed in agri-food and animal feed, pharmaceutical, and cosmetics industries. Therefore, this review aims to promote cereal bran waste and by-products, highlighting how to use them as functional ingredients with health-promoting properties and desirable technological aspects. Currently, there are few data on the nutritional exploration of these by-products as health-promoting agri-food products. Cereal bran is a nutritious natural agricultural by-product, but its potential application in the food industry is still limited due to a lack of literature focused on its quality attributes, which may become useful for informal explanation and evaluation during food product formulation. With the growing demand for fiber-rich foods, cereal bran valorization can generate revenue for milling industries.

**Keywords:** bioeconomy; sustainability; biowaste; bioactive compounds; phytochemicals; health effects; agricultural by-products; cereal bran

## 1. Introduction

Waste is directly linked to human development [1] and, often, wastes are emitted into the atmosphere without proper disposal procedures, which can contaminate the ecosystems and have serious adverse effects on human health. The improvement of waste management is becoming extremely valuable in agri-food-processing industries. In this context, the goal is to maximize the use of raw materials, which will further reduce emissions and waste management problems [2]. Food is an essential human need, and one of the major challenges faced by a large proportion of developing countries is producing enough food for the growing population [1]. Nevertheless, the issue is to improve the global food production system on a sustainable basis to increase food security and food safety without compromising the environment, thus making the idea of sustainable functional food suitable for consumers [3]. It is also significant in the production of renewable energy, biofuels, biochemicals, and biopolymers (such as polysaccharides-xylan, xanthan gum, pectin, etc.) via the bioconversion of agricultural wastes [4]. One of the major challenges faced by the developing world is to increase agricultural productivity without compromising the environment and securing the sustainable development of the food industry.

Asian cereal waste accounts for 55% of global cereal waste. Surprisingly, most of these wastes—such as straws, husks, bran, and hard shells (an inedible portion of cereals)—are simply eliminated as waste [5]. Upon examining the food matrix discarded as waste, the probability of its reuse in the food chain needs to be evaluated and restrictions should be imposed. For this purpose, the use of the term "by-product" refers to certain wastes that are substitutes for the improvement of food products, thus bran can be used to extract functional compounds that can be further utilized in numerous products, contributing to its added value and making it a more economically sustainable by-product.

The currently used milling processes of cereals generate two main by-products, bran and germ [6]. Among these, cereal bran contains health-promoting bioactive compounds such as vitamins, minerals, dietary fibers, phenolics, and flavonoids. Cereal bran should be valorized by its incorporation, as a value-added ingredient, into a variety of functional foods [7].

Recent studies have shown that antioxidants extracted from natural resources are mostly utilized as functional ingredients in food product development. The efficient use of raw material residues, food wastes, and alternate sources of food for human consumption would contribute to an improved nutritional value of the developed products derived from existing resources. Food processing waste is the end product of different food-processing sectors, which may be used for the extraction of bioactive compounds or may be directly used with some modifications for different purposes. During food processing, raw materials (cereals, fruits, and vegetables) are processed into final products, and the remaining parts represent the food industry's huge amount of waste materials [8]. Therefore, these materials are discarded as waste, thus contributing to environmental pollution. However, there are sufficient methodologies to recover them, and if the value of the waste exceeds the cost of reprocessing, these wastes may be considered added-value products. Sustainability presents the agriculture sector with both opportunities and challenges. It is an incentive due to the possibility of using by-products from food processing to extract bioactive compounds and nutrients and, simultaneously, provide enormous potential for waste minimization and indirect incomes [9]. The current review presents and discusses the actual application of industrial cereal waste materials as good sources to obtain further valuable functional compounds. So far, the most accessible research on the evaluation of cereal-milled bran investigates particular by-products and their utilization in different foods. This review aims at promoting the waste and by-products from cereal industries and highlighting the use of those as functional ingredients with health-promoting and technological properties. Currently, limited data are available on the nutritional potential of these by-products as health-promoting foods. Cereal bran is a nutritious by-product, but due to the lack of literature on its quality attributes for informal explanation and evaluation during the

formulation of food products, its potential to be used in food industries is still incipient. Chemical, nutritional, and technological aspects of cereal bran are presented in this review. With the increasing demand for fiber-enriched foods, cereal bran valorization can generate additional income for the milling industry. Cereal bran is rich in dietary fiber, which can stimulate the gastrointestinal tract and thus counteract constipation and other health effects. Cereal bran plays an important role in environmental sustainability. Environmental sustainability is a condition of balance, resilience, and interdependency, which enables actualizing the needs of human society without compromising biological diversity or outreaching the capacity of its load-bearing ecosystems to replenish the utilities required to satisfy those needs. The key feature is to define and address a common holistic strategy to manage the universal environmental risks of cereal bran and other waste materials that are locally or conjointly shared and foster resilience across states to advance inclusive and sustainable development [10]. Previous literature presented the uses and nutritional profile of cereals bran. In the current review article, we describe the physical, chemical, microbial, functional, and sensorial properties of cereal bran by-products and their components. The health benefits of cereal bran as dietary fiber are also highlighted.

## 2. Cereal Processing By-Products

In most countries worldwide, the fast-growing market of cereal-milling industries generates various valuable by-products such as straw, bran, germ, cob, etc. In general, milling industries generate a huge amount of waste and milling by-products annually. In most cases, the waste is untreated and underused and is, therefore, discarded in maximum estimates by burning, dumping, or unplanned landfilling [11]. Many industrial by-products are generated during food processing and represent a potential source of bioactive compounds with functional properties such as dietary fibers. The production of natural bioactive products has increased, while research toward alternative sources is gaining more interest. With the increasing interest in nutritious functional foods, it is argued that by-products may have a very high content of bioactive compounds [11,12]. The retrieval of compounds with significant added value allows them to be used as additives to agri-food and functional ingredients. The current waste management strategy is not only expensive but also harmful to one's health. The ability to convert by-products into edible products, such as valuable biochemical recovery, is a prerequisite for developing and optimizing environmentally friendly techniques. Agricultural waste products have a significant ecological impact, making their use an environmentally beneficial practice [13]. Continuous economic growth can also be achieved through the valorization of low-value by-products. The use of cereal milling waste to obtain by-product bio-extracts may be based on the development of novel products with bioactive properties, as well as the reduction in the environmental impact caused by disposal [14].

The food production industry has been growing very rapidly across the globe in recent years and is continuing along this path. In addition, as analytical capacity grows, more experience and knowledge are gained concerning the functions of bioactive compounds and the biochemical composition of various food products regarding their impact on human health [11]. The rapidly expanding food-processing industry in developing countries is expected to generate more waste material products in the coming years. Straw, bran, germ, cob, and husk are the most common cereal waste materials. These by-products are rich in both dietary fiber and phenolic acids. Many functional food product manufacturers have generated direct profits and significant waste reductions by using the food industry's by-products, primarily as a source of functional fiber and phenolic compounds [11]. Cereals for animal feed are processed to eliminate the germ and bran. This method depletes grains of important components that are good for human health, such as dietary fibers, minerals, vitamins, and phenolics, which have also been investigated as prospective feedstocks [15]. New techniques for cereal processing introduce more effective grain processing to generate a variety of high-value products such as biochemicals and bioenergy [16]. In the supply chain of cereal grains to final products, milling is a very important part; however, the

importance of grain for the non-food sector should not be underestimated. Wet and dry milling are the two main methods used in the grain industry, and each has certain advantages and disadvantages. Dry milling is used to separate the outer fibers and the by-products of endosperm grains identified by germs (Figure 1). In the case of barley, rice, and oats, dry milling—which removes the seed covering (testa and pericarp)—is also related to pearling.

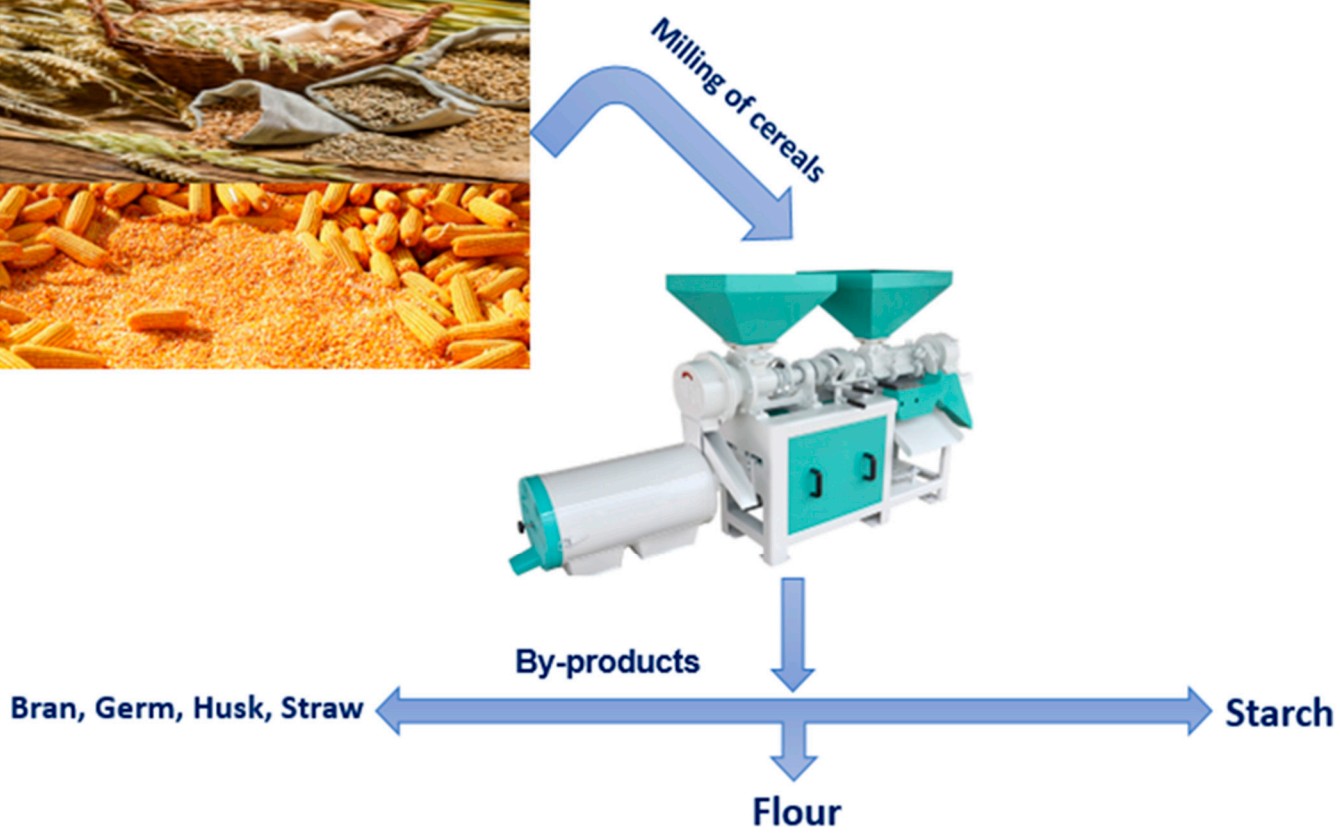

**Figure 1.** Milling of cereals, their products, and by-products.

In contrast, wet milling is primarily used to produce gluten and starch from sheer solids (rich in nutrients beneficial to pharmaceutical applications), co-product germs (projected for the oil-crushing industry), and bran, which is the combined outer layer of aleurone and pericarp. Cereal bran is a high-nutritive-quality functional ingredient with various health benefits [17]. Furthermore, bran, germ, and endosperm layers are separated from the cereal grain during milling and are considered milling by-products. In the case of the malting of grains, enzymes catalyze the conversion of starch into soluble sugars to form alcoholic beverages [18]. Moreover, cereal by-products come from both field harvesting and grain production. Most cereal waste and by-products are produced primarily during harvesting, post-harvest processing (cleaning, storage, and removal of husk materials), and post-production (biopolymers and bioplastics, biofertilizers and biofuel, bran, and germ). The main by-product produced during primary crop harvesting is straw, with this being the key raw material for biofuel production/biogas. Further, corncob is discarded in the field during maize harvesting as a result of the corn stover (mixing leaves, stalks, husks, and cob). Dry cereal milling may include flour production, wet milling for starch and glucose production, and grain brewing [19].

It is no longer enough for product development to meet technical and economic requirements; it is also important to note that their features are profitable. In this regard, these requirements help to motivate academic and industrial efforts to determine innovative and emerging technologies. Indeed, the huge number of technical documents makes

it possible to explore the research efforts regarding the application of industrial cereal waste as raw materials in composites, nanomaterials, biofuels, and energy production [20]. According to the current research, various waste forms, including the peel of pomegranates, lemons, oranges, etc., and other waste forms such as walnut husks can be used as natural antimicrobial agents [10]. Researchers such as Pasha et al. [21] have described the functional properties of industrial waste wheat straw very well.

## 3. Cereal's Bran

Cereal bran has functional components with high nutritional quality and immense nutritional characteristics. Despite its nutritional benefits, cereal bran is underutilized in food processing [22]. Parts of the starchy endosperm, pericarp, aleurone, testa, and germ contain fractions of cereal bran. The most widely consumed main cereals are sorghum, millet, barley, maize, rye, rice, oat, and wheat [23]. The dry matter of the kernel is approximately 3 to 30% bran, depending on the cereal crop type [24]. Cereal bran composition varies depending on crop type, kernel size, shape, bran layer thickness, grain storage state, and storage period prior to milling [25]. Vitamins (particularly vitamins B and E), bioactive compounds (ferulic acid, p-coumaric acid, and syringic acid), dietary fiber, and minerals are found in higher concentrations in cereal bran [26]. Protein, ash, fat, phosphorus, and niacin, the main components of bran, are all present in relatively high amounts. Cereal bran provides attributes such as thickening, emulsification gelling, and stabilization [17]. Furthermore, it has a variety of health and nutritional benefits. Global dietary guidelines recommend nutrient-dense components such as whole grains or wheat bran [27]. Cereal bran also has the commercial advantage of being used in food production as a source of dietary fiber without incurring the additional cost of dietary fiber extraction [28].

## 4. Sustainability of Cereal Bran

The term "Bioeconomy" refers to the application of biomaterials in bio-refineries employing non-food plants and trees for industrial purposes to sustainably produce and convert biomass for a variety of food, health, fibers, industrial, and energy products. Furthermore, the current management practices that discharge cereal-processing by-products into the environment are not sustainable. The reuse and valorization of their by-products is a major problem for the sustainable development of the agriculture and food sectors. Cereal by-products are a significant source of carbohydrates, proteins, lipids, vitamins, primarily B-complex and vitamin E, and inorganic and trace elements. Cereal bran is a rich and low-cost source of phytochemicals that may have use in nutraceuticals and pharmaceuticals. In the recent era, bran is increasingly added to food products because of its nutritional profile and physiological effects. Furthermore, different methods have also been suggested to improve sustainability recovery approaches in cereal by-products.

In the current review paper, efforts have been made to clarify various sustainable cereal residue management techniques in cereal systems in order to assist the development of a sustainable and economically viable advanced biofuels and bio-products sector.

## 5. Cereal's Bran Properties

### 5.1. Physical Properties

The physical properties of cereal bran show that bran as a by-product is used to improve quality characteristics and safety aspects of bran-enriched products. Furthermore, when it comes to physicochemical parameters, the moisture content during storage is an important indicator that influences the quality of bran. The rate of development of fungus that causes the degradation of cereal bran depends partially on the level of moisture content. Likewise, there will be no growth of fungi at low moisture content. Fungal growth begins at approximately 14% moisture content or slightly above [29]. Nonetheless, due to the relatively high-fat content, moisture plays a critical role in cereal bran's oxidative rancidity, particularly in rice bran [29]. This qualitative limitation has an impact on the utilization of cereal bran. To stabilize cereal bran, suitable methods employing particular

nutritional inhibitors and inactivating enzymes could be used. Wet extrusion, dry extrusion, refrigeration, and microwave heating, as well as chemical treatments with acidic acetic and acidic calcium hydroxide hydrochloric technologies, are the most commonly used to stabilize cereal bran [30]. Microwave technology is commercially advanced and efficient, has a minimal processing time, and has a minimal impact on bran color, nutritional consistency, and functional properties [30]. In addition, bran contains essential fatty acids, ferulic acid derivatives, and tocopherols, and is a cheap source of protein [31]. The color of cereal bran is a significant physical factor in cereal grains because color influences the acceptability of food products, and as the color of bran varies between types, it indicates the presence of phytochemicals [30]. The purple color of rice bran has the highest content of anthocyanin in cereal grains, which include maize, wheat, and barley, among other blue and purple bran [32]. Bounded phenolics can be found in significant amounts in light brown rice bran [33].

Sustainable natural sources of anthocyanins, as one of the most abundant cereal bran, can be used as a common pigment (colorant) in natural colors in nutraceuticals and functional foods for health-promotion ingredients [33]. Furthermore, the use of colored cereal bran in food products may help to reduce unidentified usage adverse effects in food produced by synthetic colorants while promoting their nutritional characteristics. Cereal bran particle size is a measure of the degree to which their grains are milled. The distribution of particles is valuable on the basis of dietary fiber cereal bran possession in food processing with a critical outcome [34]. Its technical versatility is affected by bran particle size because of improvements in the bran's physicochemical, rheological, swelling, water-absorption, and fat-binding properties. The particle size of cereal bran is caused by its low water-holding capacity and high oil-binding capability [35]. Furthermore, wheat flour functionality is influenced by the bran's particle size [36]. In bran-enriched bread preparation, the size of the bran particles causes a rough texture (dark crimson color, even crust) and lowers the gas retention during proofing. Due to the negative effect of bran on wheat flour, it is important to evaluate standard particle sizes during the formulation of cereal-bran-enriched products. Previous researchers have shown that, in breadmaking with cereal-bran-enriched wheat flour, the most suitable particle size of cereal bran is 200 to 500 μm [37]. In cookie preparation, medium bran particle sizes (220 to 430 μm) improve the spread ratio, color, and nutritional and sensorial properties of cereal-bran-enriched cookies [38]. Wheat cake has the highest acceptability level of the wheat and oat bran particle size (<210 μm) [39]. Furthermore, the extraction of bioactive compounds from cereal bran does not affect particle size. Shelf life does not affect the particle size during storage of cereal-bran-enriched food products; rather, it seems that particle size affects the shelf life.

The degradation rate in wheat bran is high when the particle size is smaller, excluding finely milled bran (<0.40 mm), due to a halfway-deteriorating phenomenon demonstrated by unfractionated bran [40]. While this is dependent on the fat and moisture level of the bran, enzyme and/or microbe activity may be influenced by the bran's accessible water and oxygen-taking capabilities. After grain milling, grading cereal bran in different particle sizes is crucial before marketing or use in order to analyze their technical impacts on processed foods. This is due to the consistency of the bran variation due to various milling processes and conditions that can influence cereal bran's milled chemical composition and physical possessions. It is also useful to compare the quality of the incorporated product to bran food items from other case studies when it comes to explaining the accuracy of bran particle size during product development.

### 5.2. Chemical Compositions

Cereal bran is classified by its chemical composition, treatment, and particle size in different cereal varieties. Furthermore, rice bran has more ash fat than other cereal bran, while oat bran contains more protein than other cereal bran. The amino acid content of cereal bran proteins determines its nutritional profile. Furthermore, rice bran proteins

have more beneficial amino acids than wheat bran proteins. Unlike other cereal bran, oat bran proteins have reduced tryptophan levels in leucine protein and proline compared to corn bran. Moreover, cereal bran proteins possess a nutritious and healthy amino acid composition [41,42]. The content of carbohydrates in rice bran is higher than in other cereal bran. However, the highest value is identified in maize bran. Table 1 shows the general chemical composition of cereal bran.

**Table 1.** General composition of cereal bran.

| Bran Component | Range %, Dry-Matter (d. m.) | References |
| --- | --- | --- |
| Dietary fiber | 33.4–63.0 | [43] |
| Moisture | 8.1–12.7 | [44] |
| Ash | 3.9–8.10 | [45,46] |
| Protein | 9.60–18.6 | [46] |
| Total carbohydrates | 60.0–75.0 | [45] |
| Starch | 9.10–38.9 | [44,46] |

Cereal bran is the primary origin of cellulose, hemicellulose, and lignin as dietary fiber sources [42]. The main component that can alter digestibility, bioavailability, and nutrition is the insoluble dietary fiber from bran [17]. Corn bran has the highest amount of insoluble dietary fiber, followed by wheat and rice bran, whereas in other cereal bran, such as oat bran, the amount of insoluble dietary fiber is relatively smaller and soluble dietary fiber is higher [47]. Both soluble and insoluble functional dietary fibers involve differences in their technological and biochemical properties [42]. Moreover, cereal bran is also a good source of non-starch polysaccharides, especially arabinoxylans, and different phenolic compounds. Cereal β-glucans such as those found in oat bran are abundant in barley and oat bran, ranging from 6 to 18% (in dry matter) [48]. Furthermore, it was reported that the mineral profile of wheat bran is higher when compared to rice and oat bran, whereas the calcium content in both is higher than in maize bran. In addition, rice bran shows a higher amount of manganese, phosphorus, potassium, magnesium, and iron. In maize bran, sodium is higher compared to wheat bran, which, in turn, contains higher selenium, copper, and zinc. Compared to other cereal bran, rice bran contains a higher amount of phosphorus. Furthermore, phytic acid, as an anti-nutritional component, is observed to decrease the bioavailability of certain minerals, e.g., zinc, iron, and calcium. Since the 1990s, phytic acid has demonstrated health-promoting properties, specifically in the prevention of renal calculus, cancer, and diabetes [49]. Rice bran cereals contain more vitamins E and B than most bran cereals, making them a potential source for the human body's physiological processes and metabolic pathways. In rice bran, tocopherols have an eminent amount of Vitamin E compared to tocotrienols. Therefore, tocotrienols have higher antioxidant activity compared to tocopherols [50]. Tocotrienols and seven tool isomers, namely, $\alpha$-, $\beta$-, and $\pi$-tocotrienols (but not $\beta$-tocotrienols), are higher in rice bran, along with wheat and barley, when compared to other cereal bran [51]. Vitamin A is abundantly present in maize bran when compared to other cereal bran.

The oil contents of various cereal bran have shown that their key fatty acids are oleic, linoleic, and palmitic acids. In addition, rice bran has a higher content of squalene compared to other cereal bran oils; from rice bran, the extracted oil is rich in ÿ-oryzanols, tocotrienols, tocopherols, phytosterols, and polyphenols [52,53]. In comparison to other vegetable oils, grain oil is nutritionally superior because it contains more oryzanol, (omega-3 and omega-6) fatty acids and unsaponifiable fats [17]. Rice bran oil's potential antioxidant activity and composition, in contrast to other cereal bran oils, grant it a longer shelf-life, and its low viscosity allows for easier oil absorption during cooking and considerably reduces calories [54]. Cereal bran has a higher antioxidant capacity than other milled fractions of cereals [55]. Cereal bran exhibits antioxidant, antibacterial, and anti-diabetic activities as a result of the above-mentioned composition (Figure 2) [56]. The phytochemical contents are also presented in Table 2. P-hydroxybenzoic acid, p-coumaric acid, ferulic

acid (its derivatives), vanillic acid, feruloyl oligosaccharides, and syringic are found in wheat bran [57].

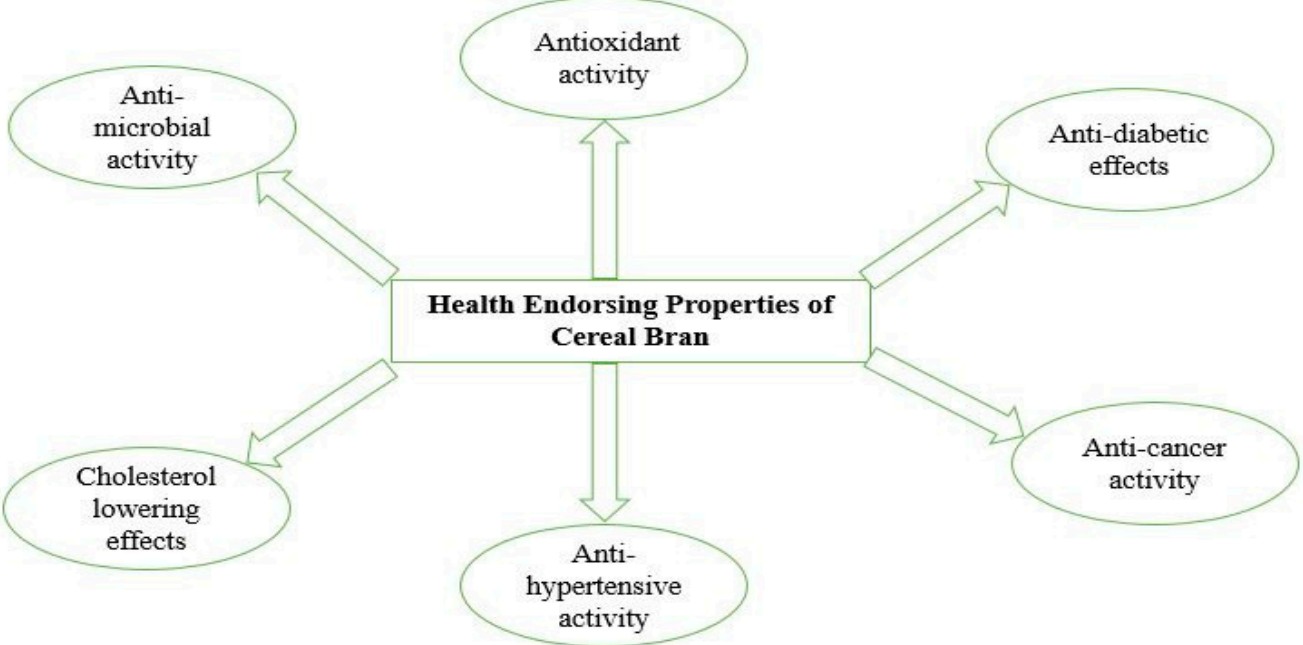

**Figure 2.** Health-endorsing properties of cereal bran.

**Table 2.** Phytochemical profile of cereal bran.

| Phytochemicals | mg·g$^{-1}$ | References |
|---|---|---|
| Alkylresorcinol | 489–1429 | [58] |
| Phytosterols | 344–2050 | [59] |
| Ferulic acid | 1376–1918 | [60,61] |
| Bound phenolic compound | 473–2020 | [60,62] |
| Flavonoids | 3000–4300 | [59,62] |

For instance, avenanthramides are present in oat bran [63]. Flavonoids, anthocyanin, and phenolic acids are found in sorghum bran [64]. Folate, steryl ferulates, phenolic acids, alkylresorcinols, and lignans are found in rice bran [65]. P-coumaric acid, diferulic acid, and ferulic acid are present in maize bran, as well as ferulated arabinoxylans and hydroxycinnamic acid conjugates [66]. The bioactive compounds in whole grains are dominant in a fraction of cereal grains such as bran [23]. Furthermore, cereal bran has increased antioxidant qualities due to the above-mentioned profiles and is used as a natural preservative in different baked, meat, and confectionery products. Moreover, as a functional ingredient containing bioactive compounds, bran can also serve as an important raw material for the development of new functional foods [67]. Due to its bioactive composition, cereal bran has been proven to prevent health complications such as non-communicable diseases (NCDs), e.g., oxidative stress, obesity, and cardiovascular diseases [68]. When applied in food product development, cereal bran functions as a barrier to the delivery of bioactive components, which is essential for optimal health. The functionality of its bioactive compounds and their health impacts on human health is illustrated in Table 3. The micronutrient profile of cereal bran is also presented in Table 4.

*5.3. Microbiological Properties*

Several studies revealed a high number of microorganisms between the pericarp and husk of cereal bran, and their metabolites may be used in further applications [68]. The use of various types of equipment during milling may increase impurities in the milling

procedure and cause the development of pathogenic microorganisms (particularly fungi and bacterial pathogens) during the cooling, fermentation, and heat treatment of mill products. A clear indication is that cereal bran fermentation may help to control indigenous microorganisms [17,67]. There are also common procedures in certain regions to eliminate microbial contaminants and divide layers of bran outer 2–4%. Using more chemicals before processing may reduce the level of chemicals in bran layers.

**Table 3.** Functions and health impacts of different components of cereal bran.

| Components | Functionality | Impact on Health | References |
|---|---|---|---|
| Dietary fiber | Increased viscosity in the gut and reduced postprandial glycemic response | The laxative effect, lowered blood cholesterol levels. Colon cancer prevention. | [46] |
| Arabinoxylans | Estrogenic effect/anti-tumor properties | Reduced risk of cardiovascular disease and type II diabetes | [62] |
| Lignin | Anticarcinogenic and antioxidant properties. Inhibition of LDL oxidation | Reduced risk of breast, and prostate cancer | [69,70] |
| Phenolic Compounds | Inhibits absorption of cholesterol in the small intestine | Reduced risk of colon cancer and cardiovascular disease | [58] |
| Phytosterols, tocopherols | Inhibits cholesterol absorption | Reduced plasma cholesterol | [50,59] |

**Table 4.** Micronutrient composition of cereal bran.

| Micronutrients | mg/100 g | References |
|---|---|---|
| Phosphorus | 900–1500 | [59,61] |
| Magnesium | 530–1030 | [61] |
| Zinc | 8.3–14.0 | [61] |
| Iron | 1.9–34.0 | [59,61] |
| Manganese | 0.9–10.1 | [59,61] |
| Vitamin E (Tocopherols/tocotrienol) | 0.13–9.5 | [61] |
| B Vitamins | | |
| Thiamin (B1) | 0.51–1.6 | [59] |
| Riboflavin (B2) | 0.20–0.80 | [59] |
| Pyridoxine (B6) | 0.30–1.30 | [61] |
| Folate (B9) | 0.088–0.80 | [59] |

## 6. Functional Characteristics of Cereal Bran

Cereal bran is involved in other innovative practical properties in food processing. The primary properties of cereal bran as a dietary fiber source are associated with its solubility, viscosity, water-binding capacity, organic-molecule-binding capability, mineral content, gel-forming ability, and oil-binding ability, affecting the end-product quality [71]. Bran has adequate capacity to absorb water and fat. The hydration properties of cereals, which are a source of dietary fiber, increase with temperature and are associated with higher dietary fiber solubility [42]. Cereal bran oil absorption is primarily related to the bran particle size [72], ranging from 0.25 to 0.59 g/mL, 148.4 to 383.7%, and 138.3 to 302.9%, regarding the bulk mass, absorption of water, and properties of cereal bran, respectively [30]. According to the aforementioned study, wheat bran has the largest capacity to bind water, fat, and water intensity, whereas barley bran is found in the uppermost bulk compactness. Cereal bran often differs in its levels of viscoelastic properties and solubility, and the food industry

could utilize its physicochemical properties in the production of novel food materials containing the building of structures, gelling, and binding water [73]. In cereal bran, the solubility and insolubility of dietary fibers depend on their biochemical impact and technical functioning [42]. Low compactness contains insoluble dietary fibers, improves intestinal microbiota development, increases fecal bulk, reduces bowel transit, regulates plasma, and decreases glycemic reactions. In addition, most insoluble fibers are fermented in the large intestine, which helps to develop intestinal microbiota, for instance, species of probiotics [73]. Because of their increased oil absorption capacity, cereal bran has great potential in the food industry. There is technical evidence that may indicate emulsions are elevated-fat foods, whereas they can modify the viscosity, water-holding capacity, and texture of formulated foods such as additives to inhibit synersis [17].

The uniqueness of cereal types has a direct impact on flour's physical and chemical properties, resulting in variations in the rheological properties of dough if used as enriched flour. The bran of cereal greatly influences the water-binding capability of dough combined with differences in water absorption produced by gluten in the production of dough [74]. These outcomes regarding the properties of the dough include increased water absorption during the addition stage, in which the consistency of the addition process affects the refusal and enhancement time; improvement in the dough during the proofing process; the dough extension (extensibility decrease) properties change; and the dough structures improve (stiffer dough or decreased stickiness) [75,76]. When compared to barley brans and oat brans, cereal bran and wheat increased dough development time (DDT) when added to wheat flour in a proportion between 0 and 40%, while the dough quality in oat and barley blends significantly decreased from 8.5 to 4 min and 7.0 to 3.5 min, respectively.

The extension resistance for formulations gradually decreased as oat and barley bran increased, whereas up to 40% of wheat bran (3.11 to 10.00), rice bran (3.11 to 18.09), and barley bran (3.11 to 8.88) were used [77]. Another study found that incorporating wheat bran into baking products increased water absorption and decreased dough stability, extensibility, and peak viscosity. Baked products' textures may be altered in addition to their rheological properties by adding cereal bran, as has been reported by previous studies [75]. In baking products, the incorporation of bran in wheat flour showed various technological, nutritional, and functional aspects. Technically, cereal bran influences the crumb structure, texture, and loaf volume [67]. Furthermore, cereal bran contains non-starch polysaccharides, which affect the development of a proper, strong gluten network, leading to rheological attributes and aggravation, as does the bread dough treatment, resulting in a hard crumb, bitter flavor, lower loaf volume, and dark color [17]. The composition and physicochemical characteristics of arabinoxylans influence their functionality [73]. In the production of bread, negative effects are induced due to unextractable arabinoxylans, while water-extractable arabinoxylans in wheat flour have positive effects on bread characteristics, including the texture of crumbs and the amount of bread. Furthermore, the dilution of gluten proteins and particle sizes can improve the final product and dough characteristics because the surface area and amount of ferulic acid increase, and the interaction between active components and gluten are also enhanced. Therefore, this impacts gluten development [75]. The particle size of the bran is affected by the functioning of the dough. There are often conflicting reports about the effect of bran fractions on dough formation in various research. For dough mixing, reduced resistance regarding the particle size of fine wheat bran has been reported [78], and as evaluated by a farinographic study, the necessity for dough mixing was reduced, associated with coarse bran. The consumption of fractions observed, according to Noort et al. [75], has the least negative impact on coarse bran. These results evaluate the impact of a particular size, which could be partly related to variability in bran description, physical uniqueness and bran structure, natural differences, or differences in the bread-baking procedure. The technological influence of cereal bran in food production has been enhanced by heat treatment and fermentation, particularly in cereal-based products [79]. The surface of bran roughness also impacted the rheological properties of dough but did not adversely impact the quality of bread preparation [80]. Wheat bran and rye were

prepared using bioprocessing techniques such as fermentation with specialized specific yeasts and enzymatic processes [27]. The use of enzymes and microbes has enhanced bran technology in the baking industry; bran in wheat flour increases the loaf volume, improves shelf life, and improves crumb quality [67,81]. Regarding dough, carbon dioxide retention was enhanced by using fermented bran, and it was reported that the utilization of enzymes enhanced the dough's stability. Amylolytic- and phytate-degrading enzymes associated with cereal bran treatment, according to Coda et al. [82], boosted the nutritional and technical consistency of bread. During bread preparation using wheat bran, the micronization technique improved technological properties [83]. The external bran division is influenced by coarse fiber parts. Pearling enhanced the nutritional consistency of wheat bran by exclusively modifying the functional properties with 10% enrichment [17]. Before adding cereal bran to the dough, the quantity of the loaf increased, and the quality of the bread improved by incorporating 12% bran in wheat bread [84]. The maize bran particle size is significantly decreased, and it may be micronized as well. Removing the microstructure results in a significant reduction in the bran matrix and an increase in the surface area [85]. Oil-holding, swelling, and water-holding capabilities, as well as cation exchange capability, were all increased by 140% and 90%, respectively. The usage of cereal bran in processing operations will be improved as a result of these improvements during food processing. The functional and technological aspects of cereal bran are shown in Figure 3.

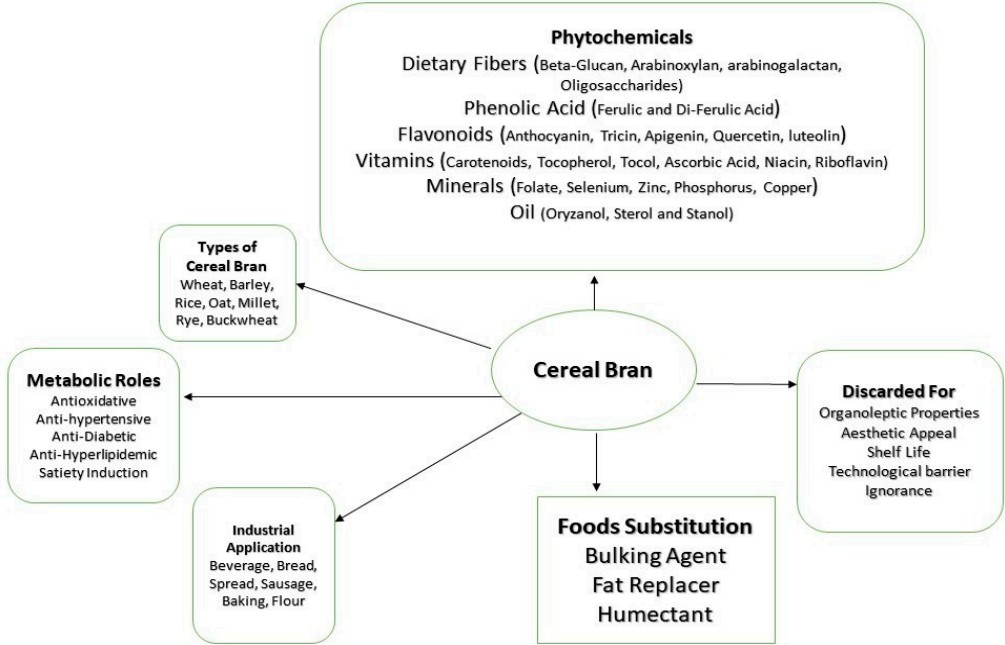

**Figure 3.** Cereal bran as a functional product.

## 7. Sensory Properties

Sensory aspects of foods are determined by the cereal's bran based on the level of bran added, bran particle size, bran treatment, texture, taste, flavor, and color. The cereal bran contains cellulose and lignin, which have a negative impact on the taste and flavor of a variety of consumer foods [67]. Mouthfeel and taste both change due to elevated fiber components, which results in a gritty texture at higher substitution levels. Studies have shown that high-fiber bakery products are not well recognized because of their effect on the technical characteristics of the final-product quality [75]. Previous research by Hussain et al. [86] shows that bran-enriched bread may reduce the loaf volume of the baked product and has relatively low acceptability scores, enhanced crumb hardness, crispness loss, changes in appearance (color and surface characteristics), and flavor. Because of their higher dietary fiber and antioxidant content, the consumer market shows potential for

cereal bran products [73,87]. Several more changes in the methods of processing cereal bran regarding the sensory qualities of foods to address their undesirable properties have been researched. Regarding bread, to ensure the most excellent sensory attributes, pre-ferment bran sourdough bran has been recognized [88]. Various kinds of research reported that wheat bran, as well as rye fermentation, is an important pretreatment process for enhancing the sensory attributes of the bran in bread. The enzyme's combined activity has been proven in yeast with bran fermentation (for example, amylase, xylanase, and lipase) to soften the bread texture [65]. Introducing fermentation with sourdough, in addition to enhancing texture properties, improved the nutritional content product quality of the final product (phenolic acids especially) [67], while the protein content, via proteolysis from cereal bran flour, also activated bioactive peptides when lactic acid bacteria (LAB) was utilized [81]. The production of bioactive components may suggest cereal bran as the best raw material, with the lactic acid bacteria bran fermentation as evidence. Bread quality can be improved by wheat bran extrusion but without negative impacts on the taste of wheat-based bread products (up to 20% wheat bran) [89]. Wheat bran's sensory qualities improved micronization in the breadmaking process [83]. With enhanced appearance and color, the bitter flavor is reduced by bran bleaching [17]. Furthermore, improvements in the processing of cereals bran must be made to maintain consumer acceptance in terms of sensory properties, food production, functional attributes, and technological factors until an acceptable level of cereal bran is achieved [90].

## 8. Potential Applications of Cereal Bran

Cereal bran is used as animal feed and poultry feed. Following thermal treatments, cereal bran has been used as a functional ingredient in different baked, meat, and confectionery products [91]. Cereal bran is also used as a fiber source and fat-reducing agent in low-calorie food products. Recently, many researchers extracted bioactive compounds from cereal bran. These bioactive compounds are used in the food, medicinal, and cosmetics industries. The main functional moieties are non-starch polysaccharides extracted from cereal bran. These moieties, especially arabinoxylans, have many functional and nutraceutical properties. Arabinoxylans are used as stable food gels and food thickness and stabilizing agents because of their water-holding capacity and highly viscous properties [92]. The biological properties of arabinoxylans, such as antioxidant, anticancer, and prebiotic activities, suggest their use as bioactive compounds in pharmaceutical and herbal medicinal industries. In previous studies, many researchers reported that cereal bran had different phenolic compounds, including ferulic acid, p-coumaric acid, syringic acid, vanillic acid, and caffeic acid. These phenolic compounds act as antimicrobial agents and antioxidants, and their further application in developing smart biogenic coatings and films and packaging has been researched [93].

In the context of the circular bioeconomy, cereal bran serves as a valuable substrate source for a variety of microorganisms that produce additional, high-quality, bio-based compounds for various industrial applications, such as xanthan gum (by *Xanthomonas campestris*) [94,95], vanillin (engineered by *E. coli*) [96], inulinase (by *Saccharomyces* sp.) [97], etc. Growing *Xanthomonas* bacteria on a medium containing a carbohydrate supply in the form of a hydrohyzate obtained from the nearly full hydrolysis of a cereal grain produces *Xanthomonas* hydrophilic colloid [95,98] for texture, coatings, and films applications. Furthermore, cereal bran has already been proven in previous literature to be a good source of dietary fiber and can be utilized in different functional and therapeutic food products to reduce financial and disease burdens [99].

## 9. Conclusions and Further Perspectives

The bioeconomy is seen, among other things, as an important driver of economic growth, and is likely to open up long-term growth potential as it offers new opportunities for adding value to renewable raw materials. Reprocessing and recovering by-products from food waste have become critical problems to investigate and illuminate nutraceutical

distinctiveness and the benefits of industrial cereal waste. As a natural resource, cereal agro-industrial waste may be useful to diminish the effect of emanation with lower production costs. Cereal bran is a natural product and provides a high nutritional value. Cereal bran is extremely rich in bioactive compounds such as carbohydrates, fats, and crude protein, as well as other health-beneficial compounds. By-products from the food-processing industry could be used directly, but with certain modifications, they could also aid in the purification and isolation of other components. Cereal bran can be particularly beneficial to health and can be used in a variety of ways. Agriculture wastes are utilized in the production of biofuels, and cereal bran might be utilized in different ways as vitamins, antibiotics, antioxidants, and enzymes. The bioeconomy provides an opportunity to create innovative bio-based products and processes, thereby opening up new markets. Furthermore, cereal bran represents a valuable substrate source for various microorganisms to produce further high-quality, bio-based compounds for different industrial applications in the context of a circular bioeconomy, e.g., xanthan gum (by *Xanthomonas campestris)*, vanillin (engineered by *E. coli*), inulinase (by *Saccharomyces* spp.), etc. It can be also used as animal feed. In the food-processing industry, phenolic compounds or non-starch polysaccharides, as isolated bioactive compounds, are used as useful ingredients in nutraceutical production or pharmaceuticals and cosmetics industries.

**Author Contributions:** Conceptualization, T.T.; methodology, T.T.; software, M.H.; validation, S.B.; formal analysis, M.; investigation, M.N.; resources, J.M.R.; data curation, M.H.; writing—original draft preparation, T.T.; writing—review and editing, H.B.U.A., A.V.R. and M.T.; visualization, S.B.; supervision, F.S.; project administration, R.M.A. All authors have read and agreed to the published version of the manuscript.

**Funding:** This research received no external funding.

**Institutional Review Board Statement:** Not applicable.

**Informed Consent Statement:** Not applicable.

**Data Availability Statement:** Not applicable.

**Acknowledgments:** The authors gratefully acknowledge the University of Lahore and Government College University, Faisalabad, Pakistan, for providing facilities regarding data collection. This work is based on the work from COST Action 18101 SOURDOMICS—Sourdough biotechnology network towards novel, healthier, and sustainable food and bioprocesses (https://sourdomics.com/; https://www.cost.eu/actions/CA18101/, accessed on 5 October 2022), where the author M.T. and A.V.R. are members, and the author J.M.R. is the Chair and Grant Holder Scientific Representative and is supported by COST (European Cooperation in Science and Technology) (https://www.cost.eu/, accessed on 5 October 2022). COST is a funding agency for research and innovation networks. Regarding the author J.M.R., this work was also financially supported by LA/P/0045/2020 (ALiCE) and UIDB/00511/2020-UIDP/00511/2020 (LEPABE) funded by national funds through FCT/MCTES (PIDDAC).

**Conflicts of Interest:** The authors declare no conflict of interest.

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
