# Peer review of "A Retrospective on the Innovative Sustainable Valorization of Cereal Bran in the Context of Circular Bioeconomy Innovations"

_sustainability, doi:10.3390/su142114597_

Round 1

Reviewer 1 Report

- The review discusses the properties and application of cereal bran as agricultural waste and its application in the circular bioeconomy. The authors discussed the physical, chemical, microbial, functional, and sensory properties of the wastes. 

- The review is well-written and promotes the development of sustainable agri-food systems. 

- Adding a separate section about potential applications of cereal bran wastes is recommended to increase the readability of the article. 

Author Response

Dear Reviewer 1,

thank you for your constructive comments.

- The review discusses the properties and application of cereal bran as agricultural waste and its application in the circular bioeconomy. The authors discussed the physical, chemical, microbial, functional, and sensory properties of the wastes. Answer: Thanks for your review. Our paper highlighted the mentioned properties as well as described the processing of cereal by-products.

- The review is well-written and promotes the development of sustainable agri-food systems. Answer: Thanks for your appreciation. We would like to thank to the referee for his/her close reading of our manuscript data. - Adding a separate section about potential applications of cereal bran wastes is recommended to increase the readability of the article.

Answer: Potential applications of cereal bran have now been added in section “7. Potential applications of cereals bran”. 

Reviewer 2 Report

Comments - eu

I have carefully read MS which was submitted for consideration in the “Sustainability” (MDPI), titled: “A retrospective on the innovative sustainable valorization of cereal bran in the context of circular bioeconomy innovations”. This review aims to promote the waste and by-products from cereal industries, and highlighting how to use these as functional ingredients with health-promoting properties and technological aspects.

This manuscript is in general well written, logically structured, well-illustrated and easy to understand. It also addresses a subject that is of great interest in the scientific community. The title describes the contents of the paper. The abstract is well written and encapsulates the entire study (a bit of introduction, objective, topic development and final remarks). The introduction gives a good background of the issue in question. However, in the introduction, the authors repeat the objectives of the work twice, although they describe them differently. So, I think they should remove the sentence (lines 90-92): ”The purpose of review is to discuss the actual application of industrial cereal waste materials as good source to obtain further valuable functional compounds.” The remaining points of the manuscript correctly address all the aspects proposed in the objectives supported by a considerable number of bibliographic references. The study would be even more valuable if the authors dedicated a point where we mentioned new products on the market obtained from cereal bran. 

Author Response

Dear Reviewer 2

we thank you for the constructive suggestions/recommandation.

Answers to reviewer:

I have carefully read MS which was submitted for consideration in the “Sustainability” (MDPI), titled: “A retrospective on the innovative sustainable valorization of cereal bran in the context of circular bioeconomy innovations”. This review aims to promote the waste and by-products from cereal industries, and highlighting how to use these as functional ingredients with health-promoting properties and technological aspects.

Answer: We would like to thank to the referee for his/her close reading of our manuscript data.

This manuscript is in general well written, logically structured, well-illustrated and easy to understand. It also addresses a subject that is of great interest in the scientific community. The title describes the contents of the paper. The abstract is well written and encapsulates the entire study (a bit of introduction, objective, topic development and final remarks). The introduction gives a good background of the issue in question. However, in the introduction, the authors repeat the objectives of the work twice, although they describe them differently. So, I think they should remove the sentence (lines 90-92): ”The purpose of review is to discuss the actual application of industrial cereal waste materials as good source to obtain further valuable functional compounds.” The remaining points of the manuscript correctly address all the aspects proposed in the objectives supported by a considerable number of bibliographic references. The study would be even more valuable if the authors dedicated a point where we mentioned new products on the market obtained from cereal bran.

Answer: Thanks for your appreciation. The suggestions have now been incorporated. Furthermore, we have added more data in new heading “Potential applications of cereals bran”. The provided data explored the utilization of cereals bran and their bioactive compounds in different industries.

Reviewer 3 Report

This manuscript reviewed the topic “Electro-microbiology: A retrospective on the innovative sustainable valorization of cereal bran in the context of circular bioeconomy innovations”. Overall, the research is organized. I recommend the acceptance of this work for publication once the following comments have been considered

(1)   At the end of the introduction section, authors must mention the latest review articles that have been published in this topic and clarify the main differences between the current work and the previous studies.

(2)   In section 2, the authors presented the Cereal processing by-products in detail, but they didn’t summarize these products in a figure taking into consideration the various waste forms. For the literature review articles, figures are very important for the reader as they help in reaching the idea easily.

(3)   Tables 1 and 2 can be merged into single table. Also, it is better to list the compositions for different types of cereals bran in this table from previous studies and comment on the presented values from your own point of view.

(4)   The manuscript needs improvement in technical/scientific writing. There are some typographical and grammatical errors, these needs to be removed. For example, line 52, the word “wella” must be replaced with “well”.

(5)   In subsection 4.2, the authors presented the main differences between different materials regarding their insoluble dietary fiber amount, non-starch polysaccharides, lignin, calcium, phosphorus …., etc. If the data of these components were summarized from literature for different materials and presented in a pie chart that contains more than several plots for individual components, it would be better.

(6)   Similarly, the data presented in section “5. Functional Characteristics of Cereal Brans” must be presented with one or more graphs to attract the reader and make the article clearer.

(7)   In addition to presenting only two figures for the current study, they are drawn traditionally, and there is no innovation in the way of displaying the data collected from previous studies. Authors should make more effort to collect more data and present it professionally using suitable graphs.

(8)   Part of the reference is out of date or not appropriate, please carefully reconsider the choice of recent references.

(9)   The quality of written English is not bad but can be improved.

Author Response

Dear Reviewer 3,

we thank you for your contructive suggestions/recommandations.

Please find below the answers:

This manuscript reviewed the topic “Electro-microbiology: A retrospective on the innovative sustainable valorization of cereal bran in the context of circular bioeconomy innovations”. Overall, the research is organized. I recommend the acceptance of this work for publication once the following comments have been considered

(1) At the end of the introduction section, authors must mention the latest review articles that have been published in this topic and clarify the main differences between the current work and the previous studies.

Answer: Thanks for your valuable suggestions. The suggestion has now been incorporated. The latest approach to the current manuscript has now been added at the same place.

(2) In section 2, the authors presented the Cereal processing by-products in detail, but they didn’t summarize these products in a figure taking into consideration the various waste forms. For the literature review articles, figures are very important for the reader as they help in reaching the idea easily.

Answer: The suggestion has now been incorporated. The Figure 1 has now been added, which summarize the cereals processing and their products.

(3) Tables 1 and 2 can be merged into single table. Also, it is better to list the compositions for different types of cereals bran in this table from previous studies and comment on the presented values from your own point of view.

Answer: Dear reviewer, the presented data in Table 1 and Table 2 described the range of mentioned components in cereals bran. So, no need to add one by one cereal. We believe that the presented data in Table 1 and 2 are now better.

(4) The manuscript needs improvement in technical/scientific writing. There are some typographical and grammatical errors, these needs to be removed. For example, line 52, the word “wella” must be replaced with “well”.

Answer: The current paper has now been improved. Grammatical, punctuation, typo errors have now been removed/corrected. Furthermore, authors have technically improved the manuscript.

(5) In subsection 4.2, the authors presented the main differences between different materials regarding their insoluble dietary fiber amount, non-starch polysaccharides, lignin, calcium, phosphorus …., etc. If the data of these components were summarized from literature for different materials and presented in a pie chart that contains more than several plots for individual components, it would be better.

Answer: Dear reviewer, the present data in 4.2 explored the chemical composition of cereals bran. The comparison of each component in each cereal bran is already

mentioned. In the present data we focused on components and then the comparison of their contents in cereals bran.

(6) Similarly, the data presented in section “5. Functional Characteristics of Cereal Brans” must be presented with one or more graphs to attract the reader and make the article clearer.

Answer: Dear reviewer, Figure 3 has already elaborated the functions of cereals bran. So, there is no need to add further graphs.

(7) In addition to presenting only two figures for the current study, they are drawn traditionally, and there is no innovation in the way of displaying the data collected from previous studies. Authors should make more effort to collect more data and present it professionally using suitable graphs.

Answer: Dear reviewer, we agreed to your comments and now we are added one more figure. Furthermore, more data has now been added to improve the current manuscript.

(8) Part of the reference is out of date or not appropriate, please carefully reconsider the choice of recent references.

Answer: We agreed with your concern. The latest and appropriate references have now been added to the reference list.

(9) The quality of written English is not bad but can be improved.

Answer: The English language has now been improved and edited. Therefore, we are bonded to add his name to the author list. So, we added his name in author list with the consent of all authors. 

Round 2

Reviewer 3 Report

No comments

Author Response

Response to the Editor comments

Q. I believe the English needs to be revised and improved.

Ans.

The English language has been revised and improved. Thank you.

Q. The focus of circular economy and sustainability should be further discussed in its own topic to reflect the title of the manuscript.

Ans.

Thank you very much for your suggestion to improve the quality of manuscript, The relevant literature has been added in the manuscript and highlighted in green color.